# Vascular Impairment, Muscle Atrophy, and Cognitive Decline: Critical Age-Related Conditions

**DOI:** 10.3390/biomedicines12092096

**Published:** 2024-09-13

**Authors:** Enzo Pereira de Lima, Masaru Tanaka, Caroline Barbalho Lamas, Karina Quesada, Claudia Rucco P. Detregiachi, Adriano Cressoni Araújo, Elen Landgraf Guiguer, Virgínia Maria Cavallari Strozze Catharin, Marcela Vialogo Marques de Castro, Edgar Baldi Junior, Marcelo Dib Bechara, Bruna Fidencio Rahal Ferraz, Vitor Cavallari Strozze Catharin, Lucas Fornari Laurindo, Sandra Maria Barbalho

**Affiliations:** 1Department of Biochemistry and Pharmacology, School of Medicine, University of Marília (UNIMAR), Marília 17525-902, SP, Brazildib.marcelo1@gmail.com (M.D.B.); 2HUN-REN-SZTE Neuroscience Research Group, Danube Neuroscience Research Laboratory, Hungarian Research Network, University of Szeged (HUN-REN-SZTE), Tisza Lajos Krt. 113, H-6725 Szeged, Hungary; 3Department of Gerontology, Universidade Federal de São Carlos, UFSCar, São Carlos 13565-905, SP, Brazil; 4Postgraduate Program in Structural and Functional Interactions in Rehabilitation, University of Marília (UNIMAR), Marília 17525-902, SP, Brazil; 5Department of Odontology, University of Marília (UNIMAR), Marília 17525-902, SP, Brazil; 6Research Coordination, UNIMAR Charity Hospital (HBU), University of Marília (UNIMAR), Marília 17525-902, SP, Brazil; 7Department of Biochemistry and Pharmacology, School of Medicine, Faculdade de Medicina de Marília (FAMEMA), Marília 17525-902, SP, Brazil; 8Department of Administration, Associate Degree in Hospital Management, Universidade de Marília (UNIMAR), Marília 17525-902, SP, Brazil

**Keywords:** vascular diseases, vascular atrophy, cognitive dysfunction, neurodegenerative diseases, oxidative stress, inflammation, aging, insulin resistance, nutrients, comorbidity

## Abstract

The triad of vascular impairment, muscle atrophy, and cognitive decline represents critical age-related conditions that significantly impact health. Vascular impairment disrupts blood flow, precipitating the muscle mass reduction seen in sarcopenia and the decline in neuronal function characteristic of neurodegeneration. Our limited understanding of the intricate relationships within this triad hinders accurate diagnosis and effective treatment strategies. This review analyzes the interrelated mechanisms that contribute to these conditions, with a specific focus on oxidative stress, chronic inflammation, and impaired nutrient delivery. The aim is to understand the common pathways involved and to suggest comprehensive therapeutic approaches. Vascular dysfunctions hinder the circulation of blood and the transportation of nutrients, resulting in sarcopenia characterized by muscle atrophy and weakness. Vascular dysfunction and sarcopenia have a negative impact on physical function and quality of life. Neurodegenerative diseases exhibit comparable pathophysiological mechanisms that affect cognitive and motor functions. Preventive and therapeutic approaches encompass lifestyle adjustments, addressing oxidative stress, inflammation, and integrated therapies that focus on improving vascular and muscular well-being. Better understanding of these links can refine therapeutic strategies and yield better patient outcomes. This study emphasizes the complex interplay between vascular dysfunction, muscle degeneration, and cognitive decline, highlighting the necessity for multidisciplinary treatment approaches. Advances in this domain promise improved diagnostic accuracy, more effective therapeutic options, and enhanced preventive measures, all contributing to a higher quality of life for the elderly population.

## 1. Introduction

The growing prevalence of neurodegenerative diseases among the elderly highlights a major challenge in modern medicine. These disorders, which include Alzheimer’s disease (AD), Parkinson’s disease (PD), and other types of dementia, contribute significantly to cognitive and motor decline in aging populations [1,2,3,4]. The pathogenesis of these conditions is complex, involving a combination of genetic, environmental, and molecular factors that result in progressive neuron loss and dysfunction [5,6,7]. This neurodegenerative process is heavily influenced by age-related changes like oxidative stress, mitochondrial dysfunction, and chronic inflammation [8,9,10]. Vascular impairment, muscular atrophy, and cognitive dysfunction are intricately interconnected and form a triad [11,12,13]. Vascular diseases impair blood circulation, resulting in a lack of vital nutrients and oxygen to the brain and muscles. This can lead to serious conditions like sarcopenia and neurodegeneration. Sarcopenia, a condition marked by a decline in muscle mass and function, is worsened by inadequate vascular function, which hampers the flow of blood to the muscles and the delivery of essential nutrients. Neurodegenerative diseases such as AD and PD are both affected by vascular deficiencies, which contribute to oxidative stress, inflammation, and damage to neurons [14,15,16].

Vascular impairment refers to a condition in which blood vessels restrict or hinder the flow of blood to the upper and lower extremities. Within this particular framework, certain crucial anatomical regions, such as the brain and muscular tissues, experience a lack of oxygenated blood, resulting in potential harm or injury [17,18,19]. Cerebral small vessel disease (SVD) increases the risk of stroke, cognitive impairment, and dementia [20,21,22]. This occurs because the blood vessels in the brain experience small subcortical infarcts, lacunes, enlarged perivascular spaces, microbleeds, and atrophy [23,24,25]. These circumstances cause insufficient blood circulation in the brain, leading to inadequate perfusion. Hypertension, smoking, aging, and diabetes are factors that contribute to the occurrence of SVD [26,27,28,29].

Hypertension is a significant risk factor that contributes to cognitive decline [30,31,32]. This condition disrupts the structure and functional integrity of the blood vessels in the brain [33,34,35]. Furthermore, the depletion of calcium and impairment of contractile function, along with the augmentation of the extracellular matrix, induce significant alterations in the blood supply to specific regions of the body [36,37,38]. This exacerbates vascular reactivity, leading to dilation, tortuosity, and the formation of microaneurysms, while also diminishing the blood flow to the brain [39,40,41]. Cerebral vascular injury encompasses a range of conditions that impact the structure and function of blood vessels in the brain, thereby affecting cognitive function. Among these conditions, brain infarctions without apparent symptoms, white matter hyperintensities (WMHs), microinfarctions, and microsurgeries are highlighted [42,43,44]. Furthermore, it is essential to take into account the dysfunction of the blood–brain barrier, events with interstitial fluid drainage, alterations in cerebral blood flow, and damage to myelin [45,46,47]. Image markers such as WMHs, microsangrings, microinfarcts, cortical superficial siderosis, enlarged perivascular spaces, and large infarcts are commonly employed for the accurate diagnosis of cerebral vascular lesions [48,49,50,51] (Figure 1).

Sarcopenia is a pathological condition characterized by a significant decline in muscle strength (dynapenia), mass (quantity), and function (quality) [52,53,54]. This condition can result in a decrease in motor coordination, an increased risk of bone fractures, and difficulties in performing everyday activities. It can also lead to mortality [55,56,57,58,59]. Age-related vascular alterations, such as decreased muscle perfusion, hinder the delivery of nutrients and oxygen. Consequently, the presence of inefficient blood vessels caused by arterial stiffness and arteriolosclerosis can lead to a decrease in lean muscle mass, ultimately causing sarcopenia [18,60,61,62]. Chronic inflammatory processes, oxidative stress, insulin resistance, and impaired blood flow, all resulting from endothelial dysfunction and calcification of skeletal muscle vasculature, play a crucial role in the development of sarcopenic conditions [63,64,65]. Furthermore, as individuals age, their muscles and blood vessels become less responsive to insulin, resulting in decreased microvascular blood flow [66,67,68]. This reduction in blood flow leads to a decrease in the supply of amino acids, as insulin plays a crucial role in redirecting blood flow from non-nutritive capillaries to nutritive capillaries. Additionally, insulin activates endothelial nitric oxide in the arterioles of the pre-capillary muscle, which in turn increases the surface area of the capillary for the exchange of nutrients [62,69,70,71].

Muscle is intricately connected to nerve tissue through the process of innervation. In the sarcopenic condition, there is a particular event in which the loss of nerve supply primarily affects fast muscle fibers, which then regain nerve supply from slow-twitching motor neurons [52,72]. Consequently, the number of slow-twitch fibers increases, explaining the slow movements that are seen as people age [73,74]. When there is inadequate blood flow and therefore insufficient supply of nutrients to the muscles, along with the occurrence of neurological disorders, a reciprocal relationship is formed. This indicates that sarcopenia can worsen neurological conditions and vice versa [75,76]. As a result, the degenerative condition deteriorates, causing a decrease in nerve supply, reduced ability to regenerate, and impaired functioning of mitochondria, sarcoplasm, and calcium ions in muscle fibers [77,78,79,80]. Furthermore, sarcopenia can alter the microstructure of both the parietal grey matter and white matter, resulting in decreased brain volumes either overall or in specific regions [81,82,83]. Another contributing factor to sarcopenia is the presence of increased muscle fat infiltration (MFI). This indicates a lower quality of muscle and is linked to thinner cortical thickness in specific regions of the brain, as well as a decrease in the volume of gray matter in both the brain and cerebellum. Additionally, MFI is associated with reduced muscle strength, impaired function, and an increased risk of mortality in adults [84,85,86]. Sarcopenia encompasses not only muscular pathology but also encompasses neurological alterations [13,87]. Whether the changes in nerve supply, either due to normal bodily processes or disease, contribute to the worsening of muscle strength and physical performance in sarcopenia. The changes involve the instability of the neuromuscular junction or alterations in myo-fibrous calcium homeostasis. Furthermore, sarcopenia is associated with cerebral decline, and decreased physical performance, such as reduced handgrip strength, gait speed, and the chair stand test which is used to verify particularly the quadriceps muscles [88,89,90,91,92,93].

Finally, it is crucial to consider the inseparable relationship between vascular diseases, sarcopenia, and neurodegeneration. Inadequate nutrient supply weakens arteries, leading to sarcopenia, a condition where essential alterations in the vascular body system can cause injuries in the muscular tissue. Arterial stiffness, the accumulation of fatty material and calcium in the arterial walls leading to the obstruction of blood flow, is a peripheral arterial disease that affects the blood supply to other tissues. It can also cause abdominal aortic aneurysm (AAA) and various other harmful changes in the blood vessels, which can disrupt the balance between the vascular, muscular, and cerebral environments [94]. Simultaneously, inadequate blood flow to the brain hinders the optimal growth of the nervous system, potentially resulting in neurodegenerative disorders such as AD [95,96,97,98,99].

The complex interplay of vascular disease, sarcopenia, and neurodegeneration, known as the inseparable triad, has received little attention, particularly in terms of understanding their common pathophysiological mechanisms. Vascular disorders hinder the flow of blood, leading to insufficient delivery of nutrients to both muscle and brain tissues. This exacerbates the symptoms of sarcopenia and neurodegenerative disorders such as AD and PD. While the specific impacts of these conditions are recognized, the exact biochemical and cellular processes that link them together are not completely comprehended. This review aims to fill the existing knowledge gap by investigating the role of arterial stiffness, oxidative stress, and chronic inflammation caused by vascular dysfunction in the development of muscle atrophy and cognitive decline. Additionally, it investigates the impact of sarcopenia on vascular and neurological health, exacerbating a detrimental cycle. The review seeks to clarify these mechanisms in order to emphasize the importance of integrated therapeutic strategies that focus on the triad. This approach aims to enhance the diagnosis, treatment, and prevention of age-related health issues, ultimately improving the quality of life and reducing illness and death rates among older individuals.

## 2. Vascular Diseases

The vascular network is an intricate arrangement comprising three distinct layers: intimate, medium, and adventitious. These layers possess various histological, biochemical, and functional attributes that are crucial for maintaining vascular balance and regulating the vascular response to stress or injury [100,101]. Additionally, they play a role in differentiating between different types of blood vessels [102]. Vascular diseases disrupt the structural integrity and functional capacity of blood vessels, leading to damage to the heart, brain, kidneys, muscles, and other organs [103,104]. The cells responsible for maintaining vascular homeostasis, which is the balance of blood vessels, are negatively affected by reactive oxygen species (ROS), chronic inflammation, alterations in blood flow, and metabolic factors such as elevated blood sugar levels, insulin, and certain types of lipids. On the other hand, compounds like polyphenols, amino acids, and omega-3 fatty acids can slow down the process of aging [105,106].

### 2.1. Pathophysiology and Causes of Vascular Diseases

Vascular diseases can be seen as the underlying cause of the development of ionic problems due to their promotion of a vasoconstrictor, pro-inflammatory, and pro-thrombotic environment, which leads to impaired regulation of the endothelium [107,108,109]. Thus, various anatomical regions may experience inadequate blood circulation, resulting in tissue damage and impaired growth. One possible explanation for this situation is endothelial senescence, which refers to the aging of the endothelial cells [110,111]. This aging process plays a significant role in the development of various health issues, including stroke, vascular dementia (VD), macular degeneration, obstructive sleep apnea, atherosclerosis, myocardial infarction, pulmonary hypertension, diabetes, renal failure, peripheral arterial disease, erectile dysfunction, and diabetic foot [112,113,114,115].

Cerebrovascular pathologies are strongly linked to neurological dysfunction [116,117]. Alterations in the blood–brain barrier (BBB) can contribute to or exacerbate the progression of neurodegenerative disorders [118,119]. Dysfunction in ion transport within the BBB is associated with acute brain damage and various neurological disorders, such as stroke, epilepsy, multiple sclerosis, VD, AD, and PD [120,121,122]. However, in order for this scenario to happen, certain conditions must be met. These conditions include the leakage of blood components such as fibrinogen, thrombin, albumin, and immunoglobulin G (IgG) from the cerebral capillaries, the accumulation of these components around the blood vessels, the degeneration of pericytes and endothelial cells, the breakdown of the BBB and tight junctions, and the leakage of red blood cells. All of these conditions are associated with vascular dysfunctions, which further confirms the connection between the vascular and nervous systems [123,124,125]

Coronary artery disease (CAD) is a form of vascular pathology. Atherosclerosis results in the constriction of arteries, which diminishes the circulation of blood to the brain and may as well result in transient ischemic attacks or strokes [126,127]. AAA is a vascular disease characterized by the delicate balance in rupture risk, presence of comorbidities, and intervention-related complications [128,129], which can lead to a potentially fatal rupture. Although CAD and AAA are separate conditions, they both arise from the abnormal remodeling of the blood vessel walls [130,131]. The vascular middle layer undergoes a structural alteration, mainly consisting of vascular smooth muscle cells (VSMCs), which leads to the formation of lesions and diseased blood vessels. Key features comprise the absence of the intimal layer, persistent inflammation, and the degradation of elastic fibers [132,133,134].

Vascular calcification (VC) is a type of vascular disease characterized by the accumulation of calcium phosphate complexes in the walls of arteries [135]. While VC is commonly associated with the natural aging process, it has also been linked to the development of vascular diseases such as diabetes, atherosclerosis, and chronic kidney disease [136,137,138]. Nevertheless, the vascular system is intricately linked to all other systems within the body, thereby rendering the brain and muscles susceptible to potential harm arising from this vascular condition. The pathogenic process can be caused by pro-inflammatory cytokines such as interleukin (IL)-1β, IL-6, IL-8, tumor necrosis factor alpha, and transforming growth factor beta (TGF-beta) [139]. These cytokines stimulate the differentiation of VSMCs into bone-forming cells and the formation of calcifications [140]. In addition, IL-29 plays an important role in immunomodulation as other interferons, via the activation of signaling pathways inducing the generation of inflammatory components. The atypical expression of IL-29 in VC-related disease hastened the process of VSMC osteogenic transformation and calcification in the presence of calcification medium (cap) by activating Janus kinase 2 (JAK2)/signal transducer and activation of transcription signaling 3 (STAT3) [141,142,143,144,145].

### 2.2. Effects on Blood Flow and Nutrient Delivery

Vascular dysfunction profoundly affects blood circulation and nutrient transportation through various intricate mechanisms. One key issue is the malfunction of the vascular endothelium, which impairs the synthesis of nitric oxide, an essential vasodilator responsible for regulating blood circulation and pressure. This dysfunction leads to increased vascular resistance and reduced tissue perfusion [146,147]. Chronic inflammation and oxidative stress exacerbate endothelial dysfunction, promoting the development of atherosclerosis [148,149]. Additionally, metabolic diseases like diabetes and obesity contribute to vascular impairment by inducing insulin resistance, which disrupts normal vascular function and nutrient transport [150,151]. Impaired cerebral autoregulation, often caused by metabolic and vascular disorders, reduces the brain’s ability to maintain consistent blood flow, impacting cognitive function and increasing the risk of death [152,153]. Sarcopenia results from decreased blood flow to the muscles, limiting oxygen and nutrient supply. These interdependent mechanisms highlight the importance of preserving vascular health to ensure adequate blood circulation and nutrient transportation, essential for preventing and managing conditions like sarcopenia and cognitive decline.

Vascular diseases also significantly impact blood flow and nutrient delivery, leading to a cascade of health problems. Atherosclerosis decreases blood flow and reduces the supply of oxygen and vital nutrients to tissues and organs. This can cause ischemia, where tissues experience a lack of blood supply, leading to pain and impaired function [154,155]. Reduced blood flow in the coronary arteries can result in angina or heart attacks, while in the peripheral arteries, it can cause peripheral artery disease (PAD), leading to pain and impaired mobility [156,157]. Hypertension, a prevalent vascular ailment, can gradually damage blood vessels, reducing their effectiveness in carrying blood and essential nutrients. This can affect the kidneys, leading to renal failure, or the brain, increasing the risk of strokes [158,159].

Moreover, diminished blood circulation caused by vascular disorders can impede wound healing and heighten susceptibility to infections, as tissues are deprived of sufficient nourishment and immune cells [160,161]. When the vascular system is compromised, the body’s ability to transport white blood cells to areas of injury or infection is impaired, leading to prolonged healing times and increased vulnerability to infections. Additionally, the reduced supply of nutrients and oxygen hampers cellular repair and regeneration, exacerbating tissue damage and dysfunction [162,163]. The overall impact on the body’s systems can be profound, affecting everything from physical mobility to cognitive function, underscoring the critical role of vascular health in maintaining overall well-being.

### 2.3. Consequences for Brain and Muscle Health

Vascular diseases have profound consequences for brain health, primarily through conditions such as stroke and VD [164,165]. When blood flow to the brain is restricted, as in the case of a stroke, brain cells are deprived of oxygen and essential nutrients, leading to cell death and potential loss of function. This can result in a range of neurological deficits, including paralysis, speech difficulties, and cognitive impairments, depending on the area of the brain affected. Chronic conditions like hypertension can also lead to small vessel disease in the brain, which is associated with cognitive decline and VD [20,166]. VD is the second most common form of dementia after AD and is characterized by problems with reasoning, planning, judgment, and memory [167,168]. These impacts not only affect the individual’s quality of life but also place a significant burden on caregivers and healthcare systems [169,170]. Furthermore, reduced cerebral blood flow can cause chronic conditions such as transient ischemic attacks (TIAs), which are temporary episodes of neurological dysfunction that increase the risk of major stroke [171,172].

The consequences of vascular diseases on muscle health are equally significant. Poor blood circulation due to conditions like PAD can lead to muscle pain, cramping, and weakness, particularly during physical activity [173,174]. This condition, known as claudication, results from inadequate oxygen delivery to the muscles, causing them to tire quickly and function less effectively [175]. Over time, the reduced blood flow can lead to muscle atrophy and loss of strength, further impairing mobility and overall physical health [176]. Additionally, the impaired delivery of nutrients and removal of metabolic waste products can exacerbate muscle fatigue and delay recovery from injuries. In severe cases, chronic insufficient blood flow can lead to critical limb ischemia, which may necessitate surgical intervention or even amputation. Poor vascular health can also result in chronic venous insufficiency, where blood pools in the veins, causing swelling, pain, and skin changes in the legs [177].

Moreover, diminished blood circulation caused by vascular disorders can impede the healing process of wounds and heighten the susceptibility to infections, as tissues are deprived of sufficient nourishment and immune cells [178,179]. When the vascular system is compromised, the body’s ability to transport white blood cells to areas of injury or infection is impaired, leading to prolonged healing times and increased vulnerability to infections. Additionally, the reduced supply of nutrients and oxygen hampers cellular repair and regeneration, exacerbating tissue damage and dysfunction [180,181]. The overall impact on the body’s systems can be profound, affecting everything from physical mobility to cognitive function, underscoring the critical role of vascular health in maintaining overall well-being. Vascular diseases can also contribute to the development of conditions like diabetic foot ulcers, which are difficult to heal and can lead to severe complications if not properly managed [182].

### 2.4. Current Treatments and Management Strategies 

Presently, the management of vascular diseases involves a blend of modifications in lifestyle, pharmaceutical interventions, and surgical interventions. Implementing lifestyle changes, such as adhering to a diet that promotes heart health, regularly participating in physical activity, ceasing smoking, and effectively managing stress, are essential for preventing and controlling vascular diseases [183,184]. Pharmacological interventions encompass the administration of antihypertensive medications such as Lisinopril to regulate blood pressure, statins like Atorvastatin to reduce cholesterol levels, anticoagulants like Warfarin to prevent the formation of blood clots, and antiplatelet drugs like Aspirin to enhance blood circulation [185,186,187,188]. In more severe cases, surgical procedures such as angioplasty, stenting, and bypass surgery are employed to restore sufficient blood circulation [189,190,191]. Angioplasty involves using a balloon to open narrowed arteries, stenting involves placing a stent to keep arteries open, and bypass surgery creates a new pathway for blood to flow around blocked arteries. These treatments, often used in combination, help manage symptoms, improve quality of life, and reduce the risk of severe complications such as heart attack and stroke.

Herbal compounds have demonstrated promise as adjunctive therapies for vascular diseases [192,193,194,195]. For instance, garlic (Allium sativum), rich in allicin, has been proven effective in lowering blood pressure and improving arterial elasticity [196]. Ginkgo biloba, for example, improves blood circulation and reduces oxidative stress thanks to its antioxidant properties [197]. In addition, hawthorn (Crataegus species) is employed for the treatment of cardiovascular conditions by expanding blood vessels and enhancing blood circulation [198,199]. Terminalia arjuna is known for its cardioprotective properties, which help in the treatment of heart failure and ischemic conditions [200,201,202]. These herbal remedies provide a comprehensive approach, improving the effectiveness of traditional treatments while reducing adverse effects. Ongoing research and standardization are essential for the complete integration of these natural compounds into conventional medical practice, guaranteeing their safety and effectiveness.

## 3. Sarcopenia

Sarcopenia is a degenerative and widespread condition affecting the skeletal muscles. It is characterized by a gradual decrease in muscle mass and strength, resulting in diminished physical abilities, heightened vulnerability, increased likelihood of falling, and potentially fatal consequences [203,204,205]. This condition predominantly affects older adults and significantly impacts their quality of life and independence. Sarcopenia’s pathophysiology encompasses various contributing factors. There is a rise in the apoptotic activity of myofibrils, which are the essential contractile units of muscle fibers, resulting in muscle degradation [206,207,208]. In addition, a decrease in the quantity of alpha-motor neurons, which play a crucial role in muscle contraction, also contributes to muscle weakness [209,210]. Reduced levels of anabolic hormones, such as testosterone and growth hormone, worsen muscle loss in the body due to hormonal imbalances [207,211,212]. In addition, increased concentrations of pro-inflammatory cytokines, which are molecules that transmit signals to promote inflammation, are essential in the advancement of sarcopenia [213,214,215]. The condition is primarily caused by vascular dysfunctions that hinder blood flow and nutrient delivery to muscles, resulting in energy deficiency. Understanding the various factors that contribute to sarcopenia is crucial for creating accurate diagnostic methods and effective treatment approaches to reduce its effects on the aging population [216,217].

### 3.1. Pathophysiology and Contributing Factors

Sarcopenia’s pathophysiology encompasses a multitude of metabolic disorders. Metabolic syndrome (MetS), which is defined by the accumulation of fat in the abdominal area, high blood pressure, impaired ability to regulate blood sugar levels, and abnormal levels of lipids in the blood, plays a major role [218,219]. This syndrome induces a state of chronic inflammation characterized by continuous oxidative stress, release of inflammatory cytokines, malfunction of mitochondria, and resistance to insulin [220,221]. These factors hinder the survival of cells, resulting in the death of myocytes and the loss of muscle mass [218,222]. In addition, the malfunction of the renin–angiotensin–aldosterone system worsens sarcopenia by hindering the circulation of blood and the supply of nutrients to muscles [223,224]. Cellular senescence pathways, which involve the aging and deterioration of cells, also contribute to muscle degradation [225,226]. The reduction in growth hormone diminishes anabolic processes that are crucial for the maintenance of muscle [212,227]. High levels of myostatin, a growth factor that hinders muscle growth, also have a crucial function [228,229]. Ultimately, denervation, which refers to the deprivation of nerve supply to muscles, results in muscle atrophy and the subsequent decline in muscle function [230,231]. The combination of these factors results in a complicated interaction of metabolic disruptions and inflammatory reactions that contribute to the advancement of sarcopenia [232,233]. This emphasizes the necessity for diverse therapeutic strategies to reduce muscle loss and maintain physical function [234,235,236].

The cardiovascular system undergoes physiological changes as a result of vascular aging and prolonged exposure to risk factors, such as hypertension and hyperglycemia. These changes result in an elevation in arterial rigidity [237,238,239,240]. Arterial stiffening leads to inadequate blood supply in different parts of the body, worsening hypertension and establishing a vicious cycle of vascular decline and compromised blood circulation [241,242]. Arterial stiffness varies in different parts of the arterial tree. Central arteries, such as the aorta, may undergo distinct stiffening mechanisms compared to peripheral arteries, because of differences in their structure and function [243,244]. Consequently, this difference in rigidity can lead to different levels of damage in tissues, such as nervous and muscular tissues. Arterial stiffening hampers the transportation of oxygen and nutrients, leading to localized tissue underdevelopment or degeneration. These conditions are aggravated by chronic inflammation and oxidative stress, resulting in cellular damage and a decrease in functionality. The decrease in flexibility and functioning of blood vessels disturbs the overall balance of the body, emphasizing the significance of addressing vascular health in order to avoid systemic complications linked to aging [245,246].

Milk fat globule-EGF factor 8 protein (MFG-E8), also referred to as lactadine, is a protein found on the surface of epithelial cells that plays important roles in anti-inflammatory mechanisms and the regeneration of tissues [247]. Nevertheless, the negative consequences of this are associated with the process of arterial aging and the deterioration of neuromuscular junctions [248,249]. This protein is essential in the progression of sarcopenia, specifically due to its influence on vascular functions [250]. MFG-E8 functions as a signal provider that prompts the binding of dying cells to macrophages, serving as a crucial mediator of inflammation in a range of conditions such as cardiovascular diseases, arterial dysfunctions, sarcopenia, and the disruption of neuromuscular junctions [251,252]. As sarcopenia progresses, the expression of MFG-E8 increases, which hinders the process of mitophagy by reducing the levels of important components like Parkin, PTEN-induced kinase 1 (PINK1), and microtubule-associated proteins 1A/1B light chain 3B (LC3B)-II/I ratio. The suppression of mitophagy results in cellular harm and contributes to the deterioration and feebleness of muscles. In addition, the buildup of MFG-E8 in the walls of arteries and neuromuscular junctions worsens cardiovascular diseases and sarcopenia as people age. This emphasizes its double function in both repairing tissues and contributing to disease processes.

Furthermore, it is worth noting that specific medications can induce a decline in muscle mass and strength due to adverse drug reactions. Statins not only help prevent cardiovascular disease, but also cause a variety of skeletal muscle symptoms, ranging from muscle pain to statin-induced myopathy, with or without elevated levels. The mechanism of this condition is caused by mitochondrial dysfunction and reduced levels of coenzyme Q10, which is a result of chloride antagonism at the muscle membrane. Furthermore, they enhance the activation of programmed cell death and atrophin-1 through the attachment of a prenyl group to small guanosine triphosphate (GTP)ases belonging to the Rho family. This leads to a reduction in the size of muscle fibers, an increase in the breakdown of muscle proteins, and an upregulation of myostatin expression in the muscle. The risk of developing sarcopenia, a condition characterized by the loss of skeletal muscle mass, is increased by the induction of hypoglycemia caused by diabetes control drugs. The closure of adenosine triphosphate (ATP)-sensitive potassium channels in muscle is an additional pathway that can lead to muscle atrophy through apoptosis and decrease muscle protein. Glucocorticoids ultimately hinder the function of fast contraction muscles (type II fibers), which have a high concentration of glucocorticoid receptors, as well as muscles that contain a combination of different fiber types. These substances hinder the growth of muscle proteins and promote their breakdown. Specifically, they prevent the absorption of amino acids needed for muscle protein synthesis in muscle fibers and hinder the activation of protein 1 binding of the eukaryotic translation initiation factor 4E and the ribosomal protein S6 kinase 1, which are responsible for stimulating muscle protein growth [253,254,255,256] (Figure 2).

### 3.2. Sarcopenic Obesity

Additionally, it is important to note the presence of sarcopenic obesity (SO) in the context of sarcopenia. SO is characterized by the simultaneous occurrence and worsening of sarcopenia as a result of increased adipose tissue [257,258]. This condition has recently been acknowledged by the European Society for Clinical Nutrition and Metabolism (ESPEN) and the European Association for the Study of Obesity (EASO) [259]. The clinical consequences of SO are considerably more severe than those observed in cases of sarcopenia or obesity occurring independently. This condition exhibits common underlying mechanisms with other diseases, including cancer, cardiovascular diseases, and kidney diseases [260,261,262]. These mechanisms include inflammation, oxidative stress, and insulin resistance, which are recognized as important factors in the development of this disease [263,264,265]. The screening for SO is conducted by assessing the concurrent presence of a high body mass index (BMI) or increased abdominal circumference, along with a change in body muscle composition [260,266,267].

Both obesity and sarcopenia are major contributors to the development of dementia [268,269]. Obesity increases the levels of pro-inflammatory cytokine IL-6 and negatively affects the ability of synapses to change and the formation of new neurons [270]. This ultimately leads to a decline in cognitive function. Increased levels of IL-6 interfere with the regular functioning of neurons and impede the formation of new neural connections, which are crucial for preserving cognitive functions [271]. In addition, myokines, such as irisin, which are released by muscles during physical activity, have a vital impact on neurological well-being [272,273]. Irisin governs the polarization of microglia, stimulates astrocytes, and adjusts insulin signaling and neuroinflammation in neurons, thus promoting brain health and cognitive functions [274,275,276]. Sarcopenic obesity worsens neurodegeneration and is associated with cardiovascular disease (CVD), cerebrovascular disease, diabetes, and depression [277,278,279]. This condition results in decreased physical activity, which further hampers muscle and cardiovascular health, all of which are crucial for sustaining cognitive function. Moreover, sarcopenic obesity diminishes the brain’s neuroprotective framework, increasing the risk of neurodegenerative diseases. Hence, it is imperative to tackle both obesity and sarcopenia by implementing specific interventions, such as advocating for physical activity and controlling inflammation [280]. This is essential in order to reduce their collective influence on neurodegeneration and overall well-being, ultimately improving the quality of life for those affected [269,281,282,283].

### 3.3. Relationship between Vascular Disease and Sarcopenia

Sarcopenia is frequently linked to CVDs. Studies indicate that there is a higher occurrence of sarcopenia in individuals with CVDs, suggesting a direct relationship between the two conditions [284,285]. Vascular diseases exacerbate sarcopenia by obstructing blood flow and impeding the supply of nutrients to muscle tissues, resulting in muscle atrophy and weakness [286,287]. The inadequate perfusion of muscles deprives them of essential oxygen and nutrients, leading to muscle cell apoptosis and reduced regenerative capacity [288,289]. This is particularly evident in conditions such as PAD, where reduced blood flow to the limbs accelerates muscle degradation [286]. Moreover, the inflammatory mechanisms and oxidative stress linked to vascular disorders further worsen muscle breakdown [290]. Chronic inflammation, a common feature of vascular diseases, elevates levels of pro-inflammatory cytokines such as TNF-α and IL-6, which promote muscle catabolism [214]. Oxidative stress, caused by an imbalance between the production of ROS and the body’s ability to detoxify these reactive intermediates, causes cellular damage and apoptosis in muscle cells [291,292]. Metabolic diseases like diabetes and obesity exacerbate these processes by causing insulin resistance [218,265,293]. Insulin resistance impairs muscle protein synthesis and increases protein degradation, further contributing to sarcopenia [294,295]. Thus, the interplay between vascular dysfunction, inflammation, oxidative stress, and metabolic disorders creates a vicious cycle that accelerates muscle deterioration.

Conversely, sarcopenia can accelerate the progression of CVDs by reducing levels of physical activity, which are crucial for preserving cardiovascular well-being [284,296]. Muscle weakness and fatigue associated with sarcopenia limit the ability to engage in regular physical exercise, leading to a sedentary lifestyle. This reduction in physical activity contributes to worsening cardiovascular risk factors, including obesity, hypertension, and dyslipidemia [297]. The connection between sarcopenia and vascular dysfunction is further affected by insulin resistance and chronic inflammation, resulting in the development of metabolic syndromes [218,298]. Insulin resistance not only affects glucose metabolism but also impacts lipid metabolism, leading to an increased risk of atherosclerosis and other cardiovascular conditions [299,300]. Therefore, it is essential to simultaneously address sarcopenia and vascular diseases through integrated therapeutic approaches that specifically focus on improving vascular health and muscle function [301]. This approach has the potential to improve overall well-being and decrease the mortality rate among elderly individuals. Early detection and comprehensive treatment strategies are crucial for managing both sarcopenia and vascular diseases [284,302]. Therapeutic interventions may include resistance training, nutritional supplementation, and pharmacological treatments aimed at reducing inflammation and oxidative stress [303,304,305]. Implementing lifestyle modifications, such as regular physical activity and a balanced diet, can also help mitigate the adverse effects of these conditions [306,307]. By addressing the interconnected pathways that contribute to both sarcopenia and vascular dysfunction, healthcare providers can enhance patient outcomes and quality of life, particularly in the aging population.

### 3.4. Impact on Physical Function and Quality of Life

Sarcopenia profoundly affects physical function and quality of life, particularly among the elderly [308,309]. As muscle mass diminishes, individuals experience a decrease in physical performance, which can manifest as reduced walking speed, impaired balance, and difficulty performing daily activities. This decline in physical capabilities increases the risk of falls and fractures, leading to a cycle of further inactivity and muscle deterioration. The reduction in muscle strength, a hallmark of sarcopenia, directly impacts the ability to carry out essential tasks such as climbing stairs, rising from a chair, or carrying groceries. Studies indicate that sarcopenia is significantly associated with lower physical performance, which in turn limits the ability to live independently and increases dependency on caregivers [310,311]. This loss of independence significantly impacts daily living, as older adults may struggle with personal care activities, household chores, and mobility, increasing their reliance on others and diminishing their sense of autonomy.

The decline in muscle health due to sarcopenia also has substantial implications for quality of life. Quality of life encompasses not just physical well-being, but also emotional, social, and psychological health. Sarcopenia-related impairments can lead to a sedentary lifestyle, contributing to obesity and metabolic disorders, which further degrade health status [312]. The inability to engage in social activities and hobbies due to physical limitations can result in social isolation, depression, and anxiety [313]. Additionally, the fear of falling or getting injured often prevents sarcopenic individuals from participating in physical exercise or outdoor activities, exacerbating their condition [314,315]. Research shows that health-related quality of life is significantly reduced in sarcopenic patients, with notable declines in physical functioning, vitality, and general health perception [308,316]. These psychological and social factors create a feedback loop that further diminishes the overall quality of life, as individuals may withdraw from social interactions and lose confidence in their physical abilities.

Furthermore, the economic and social burdens of sarcopenia are substantial [317,318]. Increased healthcare costs due to frequent hospitalizations, long-term care needs, and rehabilitation services place a significant financial strain on both individuals and healthcare systems. Families and caregivers also bear the emotional and physical stress of caring for sarcopenic individuals. The added responsibilities can lead to caregiver burnout, emotional distress, and reduced quality of life for the caregivers themselves. Comprehensive management strategies focusing on resistance training, nutritional interventions, and medical treatments are crucial in mitigating the impact of sarcopenia [319,320]. Early detection and targeted therapies can help maintain muscle mass and function, thereby improving physical performance and enhancing the quality of life for those affected [321]. Studies suggest that regular physical activity and strength training can help preserve muscle mass and strength, potentially delaying the onset of sarcopenia and its associated complications [322]. Ultimately, addressing sarcopenia holistically can lead to better health outcomes and reduce the societal and economic burdens associated with this condition, emphasizing the need for integrated care approaches that consider the physical, emotional, and social dimensions of health.

### 3.5. Strategies for Prevention and Treatment

Effective prevention and treatment strategies are crucial to mitigate the impact of sarcopenia. Among these strategies, exercise, nutrition, lifestyle modifications, and herbal compounds are paramount [323]. Exercise, particularly resistance training, plays a vital role in maintaining and enhancing muscle mass and strength. Resistance training, involving exercises that cause muscles to contract against an external resistance, is shown to be effective in combating sarcopenia [324]. This type of exercise stimulates muscle protein synthesis, improves neuromuscular function, and enhances muscle hypertrophy. Additionally, aerobic exercise complements resistance training by improving cardiovascular health and endurance, which can help maintain overall physical function [325,326].

Nutritional interventions are equally important in the prevention and management of sarcopenia [327,328]. Adequate protein intake is critical for muscle maintenance and repair. Older adults are often advised to consume higher levels of protein compared to younger individuals to counteract the anabolic resistance that occurs with aging [329]. Protein sources rich in essential amino acids, particularly leucine, are beneficial in promoting muscle protein synthesis. In addition to protein, other nutrients such as vitamin D, omega-3 fatty acids, and antioxidants play a role in muscle health. Vitamin D is crucial for muscle function, and its deficiency is linked to muscle weakness and falls [324,330]. Omega-3 fatty acids have anti-inflammatory properties that can help reduce muscle loss, while antioxidants combat oxidative stress, a factor contributing to muscle degeneration [331].

Lifestyle modifications, including maintaining a positive or neutral energy balance and reducing chronic inflammation, are essential for preventing sarcopenia [332]. Ensuring adequate caloric intake to meet energy demands without leading to obesity is crucial [333]. Obesity can exacerbate sarcopenia, creating a condition known as sarcopenic obesity, where excess fat mass further impairs physical function. Controlling inflammation through diet, physical activity, and possibly anti-inflammatory medications can help mitigate muscle breakdown. Maintaining intestinal diversity through a balanced diet that includes probiotics and prebiotics can also support overall health and muscle function [334,335].

Emerging therapeutic strategies offer additional avenues for managing sarcopenia. Research is exploring pharmaceutical interventions targeting the molecular pathways involved in muscle degradation and synthesis. For instance, myostatin inhibitors, which block a protein that inhibits muscle growth, are being investigated for their potential to enhance muscle mass and strength [228,336]. Hormone replacement therapies, particularly testosterone and growth hormone, are also under study for their anabolic effects on muscle tissue [337]. However, these treatments must be approached with caution due to potential side effects and the need for long-term safety data. Combining these pharmacological approaches with established exercise and nutritional strategies holds promise for a more comprehensive management of sarcopenia.

Additionally, herbal compounds have shown potential in the management of sarcopenia [338,339]. For example, ginseng and ashwagandha are known for their anti-inflammatory and muscle-strengthening properties [340,341]. Ginseng has been shown to improve muscle strength and physical performance, while ashwagandha can enhance muscle mass and reduce muscle damage [342,343]. Curcumin, the active compound in turmeric, has strong anti-inflammatory and antioxidant properties, which can help reduce muscle degradation and improve muscle health [344]. These herbal supplements can be integrated into dietary regimens to support muscle function and mitigate the effects of sarcopenia.

In summary, the prevention and treatment of sarcopenia require a multifaceted approach that includes exercise, nutrition, lifestyle modifications, and emerging therapies, including herbal compounds. Resistance training and aerobic exercise are foundational in maintaining muscle mass and strength, while adequate protein intake and other nutrients support muscle health. Lifestyle changes to manage energy balance and inflammation are also critical. Emerging pharmaceutical treatments and herbal compounds may offer additional benefits, though they require careful consideration [345,346,347]. A holistic approach to managing sarcopenia can significantly improve the physical function and quality of life of older adults, ultimately reducing the societal and economic burdens associated with this condition.

## 4. Neurodegeneration

Neurodegeneration refers to pathological conditions that primarily impact neurons. It refers to a collection of neurological disorders that have distinct clinical and pathological features, and specifically impact certain subgroups of neurons in specific regions of the central nervous system (CNS). Notable examples comprise AD, PD, amyotrophic lateral sclerosis (ALS), frontotemporal dementia (FTLD), and Huntington’s disease [348,349]. Furthermore, the vascular system plays a crucial role in numerous physiological processes within the human body. Problematic vascular mechanisms play a role in the development and advancement of diseases, confirming the inherent connection between vascular health and homeostasis [350,351]. Malfunctions in the vascular system can present as neurological disorders, for instance. Within this framework, it is feasible to ascertain that vascular health is intricately linked to neurodegenerative disorders. Reduced blood flow to the brain, commonly caused by conditions like high blood pressure and the buildup of fatty deposits in the arteries, can lead to a decline in cognitive function and the degeneration of nerve cells [352,353].

### 4.1. Pathophysiology and Risk Factors

PD is known by the loss of or reduction in dopaminergic neurons in the substantia nigra (SN) and progressive and irreversible aggregation of α-Sinuclein poorly folded in multiple brain regions [354,355]. Wild protein (WT) or mutant α-Sinuclein (a-syn) accumulates in PD to form oligomers that disrupt central cell systems causing neurodegeneration [356]. Notwithstanding, vascular Parkinsonism (VP) is a Parkinsonian syndrome that can be caused by cerebrovascular disease, and this pathology represents 4% of all patients with Parkinsonism [357,358]. Patients with VP are usually older, with worse cognitive ability and pseudobulbar incontinence or paralysis. In the development of VP, vascular disorders induce disruption of the cortical connections of the basal ganglion, which may cause dysfunctions of the cortex–striate–pallid–thalamus–cortical [359].

Regarding the relation between vascular disease and PD, both the severity and progression of cerebral SVD have been associated with incident Parkinsonism [358,360]. When SVD is present in PD, it negatively affects the clinical symptoms of PD. This includes a worsening of gait, cognition, and mood and may well be associated with an additional acceleration of the already progressive course of PD [361]. Regarding the pathological mechanisms in the interaction between SVD and PD, one of the options is the structural lesions of SVD located in strategic brain regions, for example, the basal ganglia [362]. Hypoperfusion can also occur in small brain vessels [362]. The two mechanisms mentioned above cause the generalized dysfunction of multiple brain pathways, including the disruption of dopaminergic and nondopaminergic pathways involved in the pathophysiology of motor and non-motor symptoms in Parkinsonism. In addition, it is suspected that the permeability of the BBB is increased in SVD, and with its dysfunction astrocytes can be damaged by impairing the exchange of interstitial fluids and neuronal energy supply [363,364]. In addition, the maturation of oligodendrocyte precursor cells can also cause problems that hinder the formation and repair of myelin and energy support to axons [365]. Finally, cerebral hypoperfusion can also promote the aggregation of alpha-synuclein, leading to the pathology of PD with subsequent depletion of soluble alpha-synuclein [361,366,367].

AD is characterized by slowly progressive neurodegeneration and cognitive decline, and symptoms tend to appear many years later. Contributions to vascular cognitive impairment and decline have remarkable importance. In this scenario, cerebrovascular disease occurs in almost all individuals with dementia, and vascular problems such as atherosclerosis, arteriolosclerosis, microinfarction, and amyloid angiopathy are prominent alongside markers of neurodegeneration, that is, vascular pathology has become an important risk factor for AD dementia. In addition, cerebrovascular diseases contribute to neuronal loss in the pathology of AD and amyloid protein-β (Aβ) and tau related to AD [368,369].

Some nerve changes in AD include irregular activated microglia and astrocytes, elevated levels of inflammation and oxidative stress within the regions of the lesion, as well as compromised vascular functionality. Blood vessels work as transporter centers and perform various stages in the maintenance of physiological homeostasis, including helping to regulate immune responses. In this sense, vascular dysfunction, especially problems in cerebral microcirculation, can help in the pathophysiology of AD. For example, microangiopathy, rupture, and hemorrhage cause chronic hypoperfusion and a reduction in cerebrospinal fluid (CSF), thus affecting normal blood circulation in the brain and neuronal function. In addition, they can assist in entering harmful substances, such as inflammatory factors in the nervous region, and impair the efflux of Aβ. Thus, the deposition of toxic proteins in the brain and subsequent neuronal damage occurs [370,371,372]. In addition, due to vascular dysfunction, individuals with AD have a higher susceptibility to hypoxia, leading to oxidative stress and resulting in various complications, including neuronal impairment and brain cell death. Therefore, the abnormal alterations in the blood vessels of AD disrupt the normal functioning of the brain and contribute to the progression of AD pathology (Figure 3) [373].

### 4.2. Link between Vascular Health and Neurodegeneration

Vascular health plays a crucial role in maintaining cognitive function and preventing neurodegenerative diseases. Emerging evidence suggests that cerebrovascular dysfunction is not only a contributing factor to vascular cognitive impairment but also has significant implications for primary neurodegenerative conditions such as AD, PD, and VD. Elevated blood pressure, a common vascular risk factor, has been consistently linked to age-related cognitive decline and the progression of neurodegenerative pathology underlying conditions like AD. Cerebrovascular disease can lead to cognitive impairment through multiple pathways. Chronic hypertension, for example, can cause damage to the blood–brain barrier, increase oxidative stress, and induce inflammation, all of which contribute to neuronal injury and cognitive decline. This vascular damage often precedes and accompanies the amyloid-beta plaques and tau tangles that are hallmarks of AD. Moreover, the compromised blood flow associated with vascular health issues can exacerbate neurodegeneration by depriving neurons of essential nutrients and oxygen, thereby accelerating the progression of cognitive deficits.

PD is another neurodegenerative disorder closely linked to vascular health. Vascular factors, including hypertension and diabetes, are known to exacerbate PD progression by promoting neuroinflammation and oxidative stress. These factors can further compromise the integrity of the blood–brain barrier, allowing neurotoxic substances to infiltrate the brain and accelerate neuronal damage. Consequently, improving vascular health through lifestyle modifications and medical management can have a protective effect against PD progression.

The interplay between vascular health and neurodegeneration is further illustrated by the overlapping risk factors and mechanisms underlying both conditions. Shared risk factors such as hypertension, diabetes, obesity, and smoking contribute to both vascular damage and neurodegenerative processes. Inflammation and oxidative stress, common in vascular diseases, also play significant roles in neurodegeneration. These shared pathways suggest that improving vascular health could potentially mitigate the risk of developing neurodegenerative diseases. For instance, managing blood pressure and blood sugar levels, adopting a healthy diet, and engaging in regular physical activity are strategies that can benefit both vascular and cognitive health. In addition to these risk factors, the relationship between cardiovascular risk trajectories and cognitive outcomes highlights the importance of early and sustained management of vascular health. Longitudinal studies have shown that individuals with a history of cardiovascular risk factors, such as elevated blood pressure and cholesterol levels, are more likely to experience cognitive decline and develop dementia. These findings underscore the need for proactive cardiovascular care as a means of preserving cognitive function and preventing neurodegenerative diseases. Interventions aimed at improving vascular health could delay or even prevent the onset of conditions like VD and AD, offering a dual benefit of enhancing both cardiovascular and brain health.

In summary, the link between vascular health and neurodegeneration is well-established, with vascular dysfunction contributing to the development and progression of cognitive impairment and neurodegenerative diseases. By addressing shared risk factors and implementing strategies to improve vascular health, it is possible to reduce the burden of neurodegenerative diseases and enhance overall brain health. This integrative approach highlights the importance of a holistic view in managing health, considering the interconnectedness of the body’s vascular and nervous systems. Strategies such as regular exercise, a balanced diet rich in antioxidants, and effective management of cardiovascular risk factors can significantly improve vascular and cognitive health, ultimately reducing the societal and economic burdens associated with neurodegenerative diseases [374].

### 4.3. Impact on Cognitive and Motor Functions

Neurodegenerative diseases have a profound impact on both cognitive and motor functions, leading to a wide range of disabilities that significantly affect the quality of life. These diseases, including AD, PD, ALS, and FTLD, result from the progressive loss of neurons in specific regions of the brain and nervous system. This neuronal loss disrupts essential neural pathways and processes, causing cognitive decline and motor impairments that often overlap and exacerbate each other. Cognitive impairments in neurodegenerative diseases are characterized by deficits in memory, executive function, language, and visuospatial skills [375,376].

In AD, the most common neurodegenerative disorder, cognitive decline begins with subtle memory lapses and progresses to severe impairments in thinking, reasoning, and the ability to perform daily activities. The accumulation of amyloid-beta plaques and tau tangles disrupts neural communication and leads to the death of neurons, particularly in the hippocampus and cortex, areas critical for memory and cognition. In PD, cognitive decline can manifest as difficulties with executive functions, such as planning and multitasking, alongside the hallmark motor symptoms of tremors, rigidity, and bradykinesia. Motor function impairments are another significant aspect of neurodegenerative diseases. In PD, the loss of dopaminergic neurons in the substantia nigra leads to motor symptoms such as tremors, muscle rigidity, bradykinesia (slowness of movement), and postural instability. These motor deficits severely limit mobility and increase the risk of falls and fractures. In ALS, the degeneration of motor neurons in the brain and spinal cord causes muscle weakness, atrophy, and eventually paralysis, affecting voluntary movements and respiratory function. FTLD and Huntington’s disease also involve motor dysfunctions, although these are often overshadowed by the prominent cognitive and behavioral symptoms.

The interaction between cognitive and motor impairments in neurodegenerative diseases is complex and multifaceted. Cognitive–motor interference, where cognitive tasks negatively impact motor performance and vice versa, is a common challenge for patients. This dual task interference can exacerbate functional limitations and increase the risk of accidents. For instance, individuals with PD may experience “freezing” episodes, where they temporarily lose the ability to move despite the intention to do so, often triggered by cognitive stressors or environmental changes. Similarly, gait disturbances in AD patients are linked to declines in cognitive function, particularly in attention and executive processing.

The co-occurrence of cognitive and motor symptoms in neurodegenerative diseases underscores the need for comprehensive management approaches that address both domains. Interventions such as cognitive rehabilitation, physical therapy, and pharmacological treatments aim to slow the progression of symptoms and improve quality of life. For example, cognitive training exercises can enhance executive function and memory, while resistance training and aerobic exercise can improve motor function and overall physical health. Pharmacological treatments, including cholinesterase inhibitors for AD and dopamine agonists for PD, provide symptomatic relief but do not halt disease progression. Emerging research into neuroprotective strategies and disease-modifying therapies offers hope for more effective treatments in the future.

### 4.4. Current Therapeutic Approaches

Pharmacological treatments remain a cornerstone of managing neurodegenerative diseases. In AD, cholinesterase inhibitors (e.g., donepezil and rivastigmine) and NMDA (N-methyl-D-aspartate) receptor antagonists (e.g., memantine) are used to alleviate cognitive symptoms by enhancing cholinergic function and modulating glutamatergic transmission. For PD, dopamine replacement therapy, primarily through levodopa combined with carbidopa, remains the gold standard, aiming to replenish dopamine levels in the brain. Additional medications, such as MAO-B inhibitors (e.g., selegiline) and dopamine agonists (e.g., pramipexole), help manage motor symptoms. ALS treatment often includes riluzole and edaravone, which are thought to reduce neuronal damage and oxidative stress, albeit with limited efficacy. Furthermore, extensive research is being conducted on novel targets and drug discoveries for neurological diseases [377,378,379].

Lifestyle modifications and supportive therapies also play critical roles in the management of neurodegenerative conditions. Physical therapy and exercise are particularly beneficial in maintaining motor function and mobility in PD and ALS patients. Occupational therapy helps individuals adapt to their environment and maintain independence in daily activities. Cognitive therapies and mental exercises can aid in slowing cognitive decline in AD patients. Speech therapy is essential for addressing communication difficulties in various neurodegenerative diseases, improving the quality of life and social interaction for patients.

Emerging therapeutic approaches are exploring the potential of herbal compounds in managing neurodegenerative diseases [380]. Herbal compounds, derived from medicinal plants, offer a rich source of bioactive molecules with neuroprotective properties [381,382,383,384,385]. Curcumin, a compound found in turmeric, has garnered attention for its anti-inflammatory and antioxidant properties, which are beneficial in combating neuroinflammation and oxidative stress in AD [220,386,387,388]. Studies have shown that curcumin can inhibit the aggregation of amyloid-beta plaques and tau tangles, key pathological features of AD, thereby potentially slowing disease progression. Another promising herbal compound is resveratrol, found in grapes and red wine. Resveratrol is known for its ability to activate sirtuin-1 (SIRT1), a protein that promotes cellular health and longevity. In the context of neurodegeneration, resveratrol’s neuroprotective effects are linked to its capacity to reduce oxidative damage, enhance mitochondrial function, and modulate neuroinflammation [389,390,391]. Preclinical studies suggest that resveratrol can improve cognitive function and delay the progression of neurodegenerative diseases. Ginkgo biloba, an herbal extract used traditionally in Chinese medicine, is another example of a natural compound with potential neuroprotective benefits [197,392]. *Ginkgo biloba* extracts are rich in flavonoids and terpenoids, which have antioxidant properties. Clinical trials have indicated that Ginkgo biloba can improve cognitive function and reduce symptoms in AD patients, possibly by enhancing cerebral blood flow and reducing oxidative stress.

In summary, the current therapeutic approaches to neurodegeneration involve a combination of pharmacological treatments, lifestyle modifications, and supportive therapies. The integration of herbal compounds into these strategies offers additional benefits, leveraging their natural bioactive properties to provide neuroprotection. Curcumin, resveratrol, and Ginkgo biloba are concrete examples of herbal compounds that show promise in managing neurodegenerative diseases. Continued research and clinical trials are essential to further understand their mechanisms and optimize their use in comprehensive treatment regimens, potentially improving outcomes for patients with neurodegenerative conditions.

## 5. Discussion

Vascular disease, sarcopenia, and neurodegeneration are intricately linked, forming a triad of interrelated conditions that significantly impact overall health. Figure 4 shows the main molecules involved in these processes. Vascular diseases impair blood circulation, reducing the delivery of essential nutrients and oxygen to tissues such as muscles and the brain. This nutrient deficiency leads to sarcopenia, characterized by the progressive loss of muscle mass and function, resulting in frailty, falls, and decreased physical performance. The impaired blood flow also affects the brain, contributing to neurodegenerative diseases such as AD and PD. Oxidative stress, chronic inflammation, and insulin resistance are common pathophysiological mechanisms underlying these conditions. For instance, oxidative stress and inflammation caused by vascular dysfunction can damage neurons, leading to cognitive decline and motor impairments [393]. Similarly, reduced muscle perfusion exacerbates sarcopenia, while neurodegenerative processes can further impair muscular function through disrupted neural innervation. Therefore, addressing these interconnected conditions through integrated therapeutic strategies is crucial for improving the diagnosis, treatment, and prevention of age-related health issues, ultimately enhancing the quality of life for affected individuals.

Vascular disease contributes to sarcopenia and neurodegeneration through multiple interconnected mechanisms. Vascular diseases impair blood flow, reducing the delivery of oxygen and essential nutrients to muscles and the brain. This leads to muscle atrophy, a hallmark of sarcopenia, as muscles require a consistent supply of nutrients to maintain mass and function. Vascular dysfunctions, such as atherosclerosis and arterial stiffness, promote chronic inflammation and oxidative stress, exacerbating muscle degradation and contributing to sarcopenia. In the brain, reduced blood flow and nutrient supply can lead to neuronal death and impaired synaptic function, key features of neurodegenerative diseases like AD and PD. Additionally, vascular diseases can disrupt the blood–brain barrier, allowing harmful substances to enter the brain and further damage neural tissues. Insulin resistance and MetSs associated with vascular diseases also impair muscle protein synthesis and increase muscle degradation, worsening sarcopenia. The interaction among vascular dysfunction, inflammation, oxidative stress, and metabolic disturbances emphasizes the intricate connection between vascular disease, sarcopenia, and neurodegeneration. This underscores the need for integrated therapeutic strategies to alleviate these conditions and enhance patient outcomes.

Sarcopenia and neurodegenerative processes are intricately linked, forming a detrimental cycle that exacerbates both conditions. Sarcopenia, characterized by the progressive loss of muscle mass and function, leads to physical frailty and increased fall risk, which can precipitate or worsen neurodegenerative conditions such as AD and PD. The muscle loss associated with sarcopenia reduces the production of myokines, which are crucial for maintaining neuroplasticity and cognitive function. This decrease in myokines can impair brain function and accelerate neurodegeneration. Conversely, neurodegenerative diseases contribute to sarcopenia by disrupting the neural pathways responsible for muscle innervation and function. For instance, PD, which affects motor neurons, directly impairs muscle control and contributes to muscle atrophy. AD, through mechanisms such as oxidative stress and chronic inflammation, can similarly lead to muscle deterioration. Both conditions share common pathological features such as mitochondrial dysfunction, increased oxidative stress, and chronic inflammation, creating a feedback loop that worsens both muscle and cognitive health. Addressing this bidirectional relationship is crucial for developing therapeutic strategies that target both sarcopenia and neurodegeneration simultaneously to improve outcomes for affected individuals

Significant correlations between vascular disease, sarcopenia, and neurodegeneration can be identified by examining common risk factors and physiological mechanisms. Common risk factors include the natural aging process, hypertension, diabetes, and chronic inflammation. These factors contribute to the dysfunction of the endothelium, which obstructs the circulation of blood and the transportation of nutrients to both muscles and the brain, exacerbating the conditions of sarcopenia and neurodegeneration. Vascular dysfunction leads to oxidative stress and chronic inflammation, which in turn cause cellular damage in muscles and neurons. Insulin resistance, a common feature of MetS, impedes the process of muscle protein synthesis and accelerates muscle wasting, thus contributing to sarcopenia. Simultaneously, the coexistence of impaired glucose metabolism and oxidative damage plays a role in the progression of neurodegeneration, leading to a decline in cognitive functions and motor skills. The interrelated physiological mechanisms emphasize the importance of a comprehensive approach in treating these conditions, with a specific focus on integrated therapeutic strategies that aim to improve vascular health, preserve muscle mass, and protect the nervous system. Acquiring a thorough comprehension of these interconnected mechanisms is crucial for developing effective interventions that improve quality of life and reduce the incidence of illness and mortality associated with these age-related conditions.

The interrelated nature of vascular disease, sarcopenia, and neurodegeneration has profound implications for their diagnosis, treatment, and prevention. From a diagnostic perspective, having a clear understanding of the shared pathophysiological mechanisms such as chronic inflammation, oxidative stress, and insulin resistance can improve the ability to detect and differentiate these conditions at an early stage. By employing biomarkers and advanced imaging techniques, one can obtain a thorough understanding of the degree of vascular damage, muscle atrophy, and neurodegenerative alterations. An essential aspect of treatment is the implementation of an integrated approach that specifically focuses on the triad. This includes pharmacological interventions like anti-inflammatory and antioxidant therapies, alongside lifestyle modifications such as exercise and nutritional strategies to improve vascular health, muscle mass, and cognitive function. Preventive measures concentrate on reducing common risk factors such as hypertension, diabetes, and sedentary lifestyles through public health campaigns that encourage physical activity, healthy eating, and regular medical check-ups. By addressing these interconnected conditions holistically, it is possible to slow their progression, reduce morbidity, and improve the overall quality of life for affected individuals

Future research should prioritize several key areas in order to better understand the triad of vascular disease, sarcopenia, and neurodegeneration. Initially, conducting research on the molecular and cellular mechanisms that underlie these interconnected conditions will offer a more profound understanding of their shared pathophysiological pathways. Oxidative stress, chronic inflammation, and insulin resistance should all be investigated as potential links between these diseases. In addition, the advancement of sophisticated biomarkers and imaging techniques can improve the early detection of diseases and provide more accurate monitoring of their progression. Longitudinal studies are crucial for comprehending the temporal connections and causal associations between vascular dysfunction, muscle atrophy, and cognitive decline. Furthermore, examining the impact of lifestyle interventions, such as engaging in physical activity and making dietary modifications, on mitigating the intensity of these conditions can offer pragmatic, non-pharmaceutical strategies for both preventing and managing them. The integration of multidisciplinary approaches encompassing neurology, cardiology, and gerontology will play a pivotal role in the development of comprehensive treatment plans. Exploring the possibilities of new therapeutic agents, such as anti-inflammatory drugs and antioxidants, has the potential to create new opportunities for treatment. The translation of these findings into effective clinical practices that enhance patient outcomes and quality of life will heavily rely on collaborative endeavors among research institutions and clinical settings [394,395,396,397,398,399,400].

## 6. Conclusions

This review highlights the complex interrelationships among vascular disease, sarcopenia, and neurodegeneration. These conditions share similar underlying causes, such as oxidative stress, chronic inflammation, and impaired blood flow, which collectively lead to muscle atrophy and cognitive decline. Vascular impairments are related to reduced nutrition and oxygen supply to cells. This imbalance can increase the production pf free radicals such as ROS and augment the release of pro-inflammatory cytokines such as IL-6, IL-8, TNF-α, and TGF-β. Moreover, the excessive production of IL-29 deliberately worsens vascular conditions due to the activation of JAK2 and STAT3. This scenario, associated with systemic inflammation and oxidative stress impairs muscle synapses, resulting in sarcopenia, and increasing morbidity and mortality. Understanding this triad is essential as it emphasizes the significance of a comprehensive approach to managing these interrelated conditions. Healthcare providers can develop holistic diagnostic, therapeutic, and preventive strategies by identifying the common risk factors and pathways involved. To effectively address these complexities, it is crucial to use integrated approaches that combine pharmacological interventions, lifestyle modifications, and advanced diagnostic techniques. For instance, resistance training and aerobic exercise are foundational in maintaining muscle mass and strength, while adequate protein intake and other nutrients support muscle health. In addition to traditional treatments, herbal compounds have shown potential benefits in managing these conditions. Curcumin, for example, is known for its anti-inflammatory and antioxidant properties, which can help reduce muscle degradation and improve muscle health. Similarly, Ginkgo biloba has been used to improve cognitive function and blood circulation, demonstrating potential neuroprotective benefits. Ultimately, enhancing our understanding of the connections among vascular health, muscle function, and cognitive performance will lead to improved patient outcomes, heightened quality of life, and decreased morbidity and mortality rates in older populations.

## Figures and Tables

**Figure 1 biomedicines-12-02096-f001:**
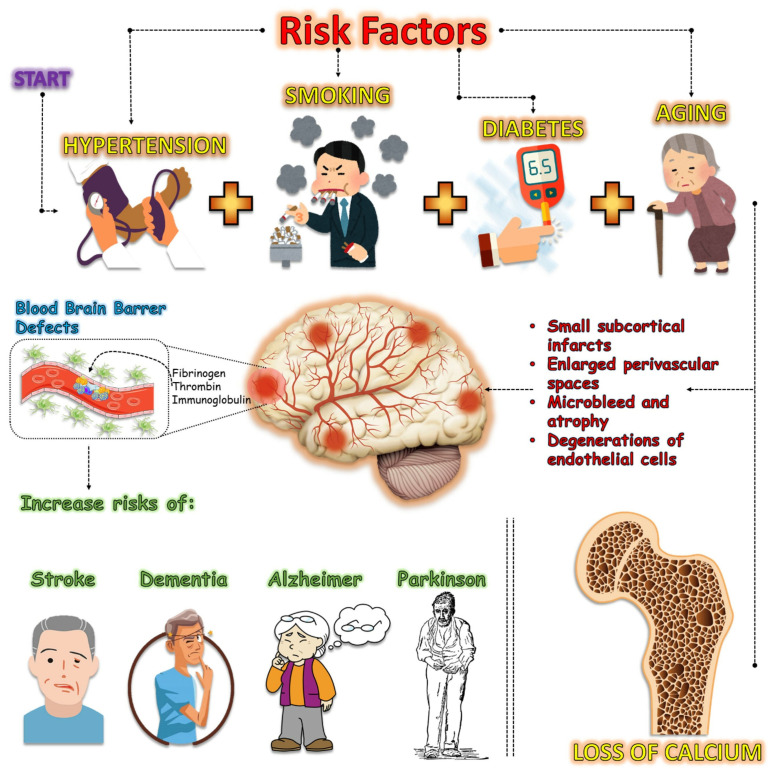
Vascular impairment and cognitive dysfunction: The vascular system serves numerous functions within the human body. When it is not functioning properly, it is associated with risk factors such as hypertension, aging, and diabetes. The vascular system possesses the capacity to expand the perivascular spaces, promote microhemorrhage and atrophy, and facilitate subcortical infarctions. In this specific scenario, especially within the brain, the blood–brain barrier may undergo a decline in functionality and be impacted by disorders in the endothelial cells, as well as the presence of defensive substances such as fibrinogen, thrombin, and immunoglobulin. The outcome is a heightened susceptibility to the occurrence of stroke, cognitive decline, dementia, Alzheimer’s disease (AD), and Parkinson’s disease (PD).

**Figure 2 biomedicines-12-02096-f002:**
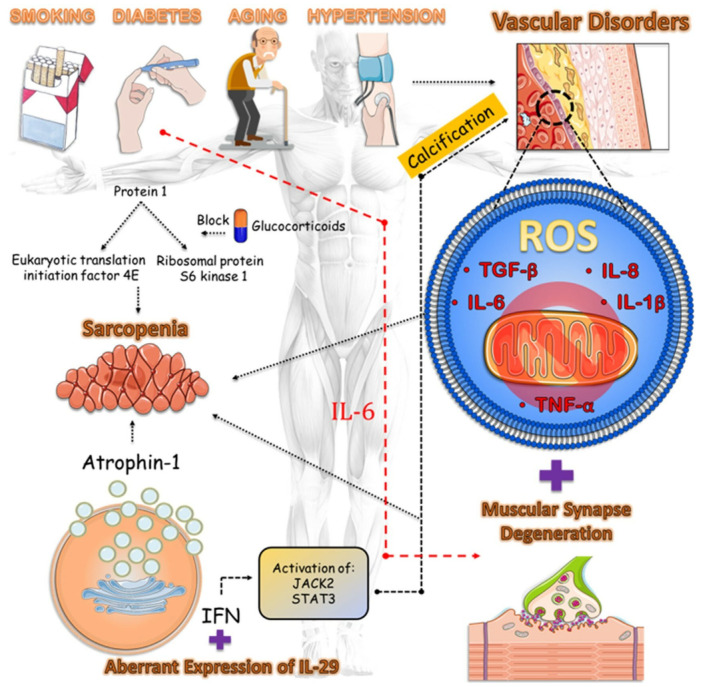
Vascular problems and neurodegeneration resulting in sarcopenia: Vascular disorders can lead to reduced nutrition and minimal oxygen supply to cells. In this context, reactive oxygen species (ROS) are formed due to energy problems in the cell, forming pro-inflammatory cytokines such as IL-6 (interleukin-6), IL-8, tumor necrosis factor alpha (TNF-α), and transforming growth factor beta (TGF-β). In addition, excess interleukin-29 (IL-29) exocytosis worsens vascular problems by activating Janus kinase 2 (JAK2) and signal transducer and activation of transcription signal 3 (STAT3). This scenario, associated with impaired muscle synapses, may result in sarcopenia, impacts on quality of life and usual activities, and increasing mortality.

**Figure 3 biomedicines-12-02096-f003:**
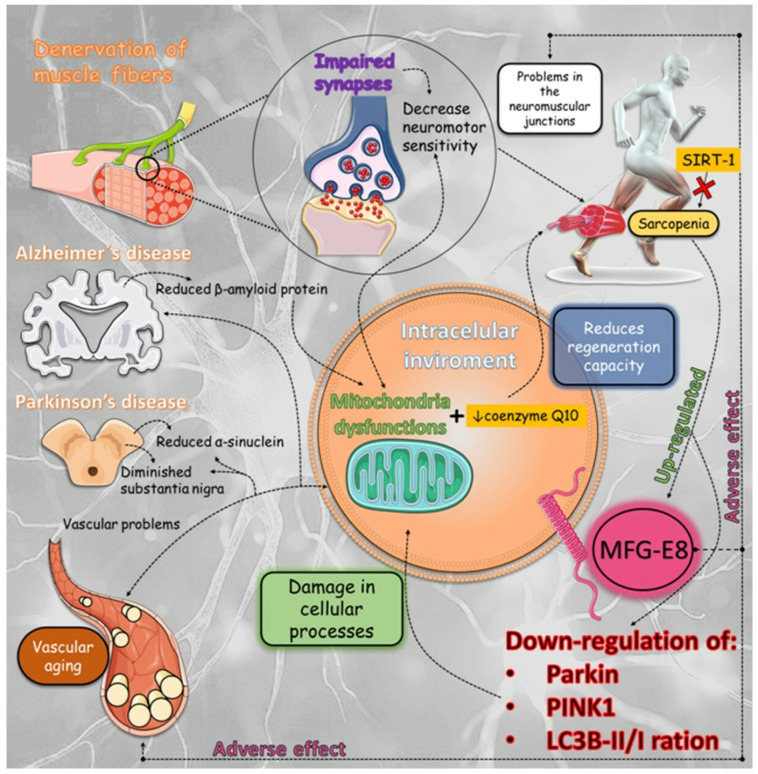
Vascular problems, neurodegeneration, and sarcopenia: There are multiple associations between the denervation of muscle fibers, vascular disease, and sarcopenia. First, the denervation of muscle fibers impairs synapses in the muscle, decreasing neuromotor sensitivity. Through this innervation loss, oxygen species occur in the cell, which can cause mitochondrial dysfunctions. This dysfunction, associated with a decreased concentration of coenzyme Q10, can impair the energy supply of the muscle fibers. In addition, cell dysfunction can promote aberrant expression of the MFG-E8 (milk fat globule-EGF factor 8 protein) cell membrane protein, increasing problems of neuromuscular junctions, mitochondrial dysfunction, and problems in the vascular system. SIRT-1 (Sirtuin-1) is a protein that promotes cellular health and longevity, and the concentration of this chemical mediator is also reduced during the sarcopenia. Finally, vascular problems can promote Parkinson’s disease by reducing nutrients from the formation of substantia nigra and α-Sinuclein, and Alzheimer’s disease by reducing nutrients for the formation of β-amyloid protein. LC3B-II/I ratio: microtubule-associated proteins 1A/1B light chain 3B-phosphatidylethanolamine conjugate/microtubule-associated proteins 1A/1B light chain 3B ratio; PINK-1: PTEN-induced kinase 1.

**Figure 4 biomedicines-12-02096-f004:**
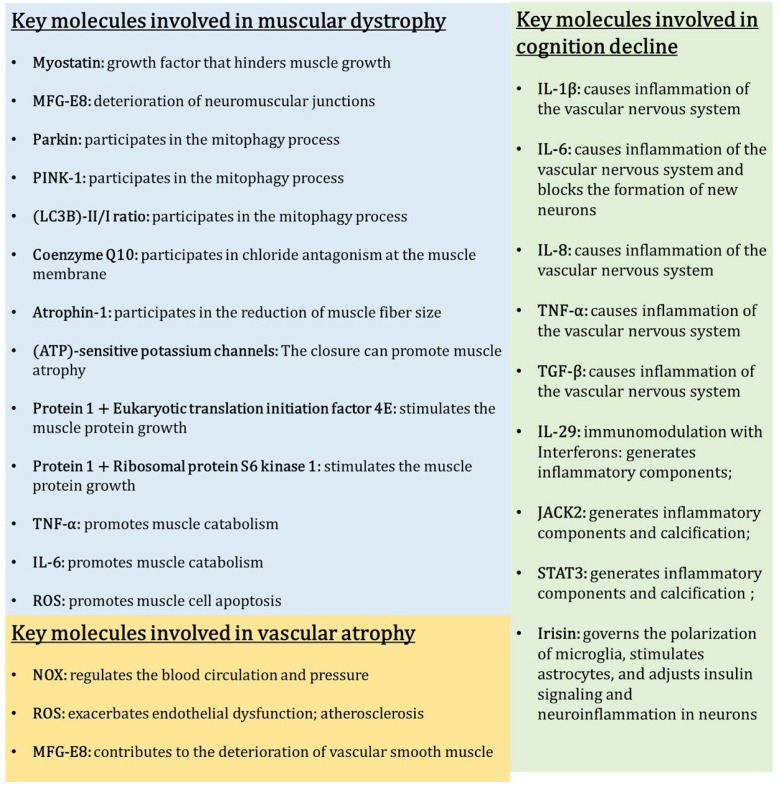
Key molecules related to muscular dystrophy and decline in cognition and vascular atrophy. ATP: adenosine triphosphate; IL: interleukin; JAK2: Janus kinase 2; LC3B-II/I ratio: microtubule-associated proteins 1A/1B light chain 3B; MFG-E8: milk fat globule-EGF factor 8 protein; PINK-1: PTEN-induced kinase 1; ROS: reactive oxygen species; STAT3: signal transducer and activation of transcription signal 3; TNF-α: tumor necrosis factor alpha; TGF-β: transforming growth factor beta.

## Data Availability

Not applicable.

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
