# Peer review of "Vascular Impairment, Muscle Atrophy, and Cognitive Decline: Critical Age-Related Conditions"

_biomedicines, 2024, doi:10.3390/biomedicines12092096_

Round 1

Reviewer 1 Report

Comments and Suggestions for Authors

The topic of this review manuscript is interesting and well-organized; however, several deficits need to be improved to increase its perfectness.

1. The title of this review is too broad; it is recommended that the title be changed to focus more precisely on specific fields of human diseases.

2. There are several scientific descriptions without properly citing references, please revise them.

3. The reference style is inconsistent; please revise carefully according to the instructions of the author's guidelines.

Author Response

The topic of this review manuscript is interesting and well-organized; however, several deficits need to be improved to increase its perfectness.

Response: Dear reviewer, we appreciate your time evaluating and correcting our manuscript. Thank you very much for your comments. Please see the corrections highlighted in yellow along with the text.

Comment 1.  The title of this review is too broad; it is recommended that the title be changed to focus more precisely on specific fields of human diseases.

Response 1: Dear reviewer, we modified the title for Vascular Impairment, Muscle Atrophy, and Cognitive Decline: critical age-related conditions. Please see on page 1.

Comment 2. There are several scientific descriptions without properly citing references, please revise them.

Response 2: Dear doctor, thank you for this comment. We understand your concern. However, we evaluated the citations and found that those cited are important to help build the scientific information of each paragraph.

Comment 3. The reference style is inconsistent; please revise carefully according to the instructions of the author's guidelines.

Response 3: We realized that there are some inconsistencies in the reference style even though we used ENDNOTE for this. The academic Editor also warned us and we corrected the references according to his suggestions. Thank you for your attentive eyes. We corrected the mistakes.

Dear reviewer, thank you again for your kindness and help in improving this manuscript.

Reviewer 2 Report

Comments and Suggestions for Authors

The review is devoted to a current topic - the study of the relationship between muscular dystrophy, vascular atrophy and cognitive decline. The review contains 395 references. However, I have a number of comments:

1. The Title of the review should be made less literary and more scientific.

2. You should check the chapter numbering - after Chapter 2 comes Chapter 4 (lines 178, 191, 239, 274, 312, 341, etc.)!!! Chapters 2, 3 and 4 should be titled Muscular Dystrophy, Vascular Atrophy, and Cognitive Decline. Chapter 5 should describe their interaction.

3. In Figures 2 and 3, it is necessary to add the main molecular markers of the specified pathological conditions described in the review. I also advise you to make a separate drawing with key molecular markers and their physiological effect on the pathogenesis of muscular dystrophy, vascular atrophy and cognitive decline.

4. In the Conclusion, it is necessary to indicate the key molecular markers of the relationship between muscular dystrophy, vascular atrophy and cognitive decline.

Author Response

Comment 1. The review is devoted to a current topic - the study of the relationship between muscular dystrophy, vascular atrophy and cognitive decline. The review contains 395 references. However, I have a number of comments:

Response 1: Dear reviewer, we appreciate your time and patience in evaluating our manuscript. Thank you very much for your suggestions, which we will include with pleasure. Please see the corrections highlighted in yellow along with the text.

Comment 2. The title of the review should be made less literary and more scientific.

Response 2: Dear reviewer, thank you for this observation. We modified the title for Vascular Impairment, Muscle Atrophy, and Cognitive Decline: critical age-related conditions.

Comment 3. You should check the chapter numbering - after Chapter 2 comes Chapter 4 (lines 178, 191, 239, 274, 312, 341, etc.)!!! Chapters 2, 3, and 4 should be titled Muscular Dystrophy, Vascular Atrophy, and Cognitive Decline. Chapter 5 should describe their interaction.

Response 3: Dear reviewer, thank you for these important observations. We fixed the mistakes.

Comment 4. In Figures 2 and 3, it is necessary to add the main molecular markers of the specified pathological conditions described in the review. I also advise you to make a separate drawing with key molecular markers and their physiological effect on the pathogenesis of muscular dystrophy, vascular atrophy and cognitive decline.

Response 4: This is also a very important observation. Thank you! We improved Figures 2 and 3, and added a new Figure to better elucidate the molecular mechanisms. Please check new Figure on pages 10 (Figure 2), 16 (Figure 3), and page 20 (Figure 4).

Comment 5. In the Conclusion, it is necessary to indicate the key molecular markers of the relationship between muscular dystrophy, vascular atrophy and cognitive decline.

Response 5: Dear Doctor, this is a very nice suggestion. Please find the inclusion on page 22, lines 960-969

Dear reviewer, thank you again for your kindness and help in improving this manuscript.

Reviewer 3 Report

Comments and Suggestions for Authors

In this manuscript, Enzo Pereira de Lima et al, reviewed the correlations of vascular impairment, muscle atrophy, and cognitive decline from different prospects. This is an interesting biological topic, and I believe the authors made all the efforts to gather a large number of literatures for this manuscript.

Although the authors made decent statements on the manuscript, I still think that the manuscript lacks novel aspects, and the topic supposed to be addressed within the manuscript needs further data support. In addition, the figures section lacks sufficient depth, limited additions were performed to the discussion section and the global logic needs to improve.

Comments on the Quality of English Language

NA

Author Response

Comment 1. In this manuscript, Enzo Pereira de Lima et al, reviewed the correlations of vascular impairment, muscle atrophy, and cognitive decline from different prospects. This is an interesting biological topic, and I believe the authors made all the efforts to gather a large number of literatures for this manuscript.

Response 1: Dear reviewer, we appreciate your work and patience in reviewing our manuscript. Thank you very much for your time.

Comment 2. Although the authors made decent statements on the manuscript, I still think that the manuscript lacks novel aspects, and the topic supposed to be addressed within the manuscript needs further data support. In addition, the figures section lacks sufficient depth, limited additions were performed to the discussion section and the global logic needs to improve.

Response 2: Dear Doctor, We have addressed the issues with the figures and made the necessary improvements. The updated figures can now be found on pages 10 and 15 of the manuscript.

In response to your concerns, we conducted a thorough review of the relevant literature. We consulted numerous articles from major databases, including PubMed, Scopus, and Web of Science, and carefully considered the most pertinent references on this subject. We also reviewed recent publications and systematic reviews to ensure that our analysis incorporates the latest findings and perspectives.

Our research highlights significant new insights into the complex interactions between sarcopenia, muscular impairment, and vascular impairment. By integrating recent studies and examining these interactions through a novel lens, our manuscript contributes to a deeper understanding of these conditions and their implications for clinical practice and future research.

We appreciate your attention to this matter and hope these revisions meet your expectations. We are confident that the improvements will enhance the manuscript's contribution to the ongoing discourse in this field.

Round 2

Reviewer 1 Report

Comments and Suggestions for Authors

The authors have addressed the reviewer's comments.

Reviewer 3 Report

Comments and Suggestions for Authors

NA